:PLOS | ONE

# Lower IQ and poorer cognitive profiles in treated perinatally HIV-infected children is irrespective of having a background of international adoption

M. Van den Hof[1]*, A. M. ter Haar[1], H. J. Scherpbier[1], P. Reiss[2,3,4], F. W. N. M. Wit[2,3,4], K. J. Oostrom[5], D. Pajkrt[1]

1 Emma Children's Hospital, Amsterdam University Medical Centers, Academic Medical Center, University of Amsterdam, Pediatric Infectious Diseases, Amsterdam, the Netherlands, 2 Department of Global Health, Amsterdam University Medical Centers, Academic Medical Center, University of Amsterdam and Amsterdam Institute for Global Health and Development, Amsterdam, the Netherlands, 3 HIV Monitoring Foundation, Amsterdam, the Netherlands, 4 Department of Internal Medicine, Division of Infectious Diseases, Amsterdam University Medical Centers, Academic Medical Center, University of Amsterdam, and Amsterdam Infection and Immunity Institute, Amsterdam, the Netherlands, 5 Emma Children's Hospital, Amsterdam University Medical Centers, Academic Medical Center, University of Amsterdam, Psychosocial Department, Amsterdam, the Netherlands

* m.vandenhof@amsterdamumc.nl

**Data Availability Statement:** All relevant data are within the manuscript and its Supporting Information files.

## Abstract

### Background

HIV-associated cognitive deficiency in perinatally HIV-infected (PHIV) children has been studied in Western countries in a population of which an increasing proportion has been internationally adopted. Studies often lack an appropriate internationally adopted HIV-uninfected control group, potentially confounding the relationship between HIV and cognitive functioning. This study aims to further elucidate the association between treated HIV infection and cognitive development by addressing the background of international adoption.

### Methods

We cross-sectionally studied the impact of HIV on cognition by comparing PHIV children and HIV- uninfected controls, matched for age-, sex-, ethnicity-, socioeconomic status (SES)- and adoption status. We used a standardized neuropsychological test battery to measure intelligence (IQ), and the cognitive domains of processing speed, working memory, executive function, learning ability and visual-motor function and compared outcomes using lineair regression models, adjusted for IQ. We determined cognitive profiles and cognitive impairment by using multivariate normative comparison (MNC) and explored associations with HIV disease- and treatment-related factors.

### Results

We enrolled fourteen PHIV children (mean age 10.45 years [1.73 SD], 93% adopted from sub-Saharan Africa at a median age of 3.3 years [IQR 2.1–4.2]) and fifteen HIV- uninfected

**Funding:** This work was supported by AIDSfonds (grant number 2015009).

**Competing interests:** The authors have declared that no competing interests exist.

controls. Groups did not clinically nor statistically differ in age, sex, ethnicity, SES, region of birth, adoption status and age at adoption. PHIV scored consistently lower on all cognitive domains and MNC outcomes. Compared to controls, PHIV children had a significant lower IQ (mean 81 [SD 11] versus mean 97 [SD 15], $p = 0.005$), and a poorer cognitive profile by MNC (Hotelling's $T^2$ mean -4.36 [SD 5.6] versus mean 0.16 [SD 4.5], $p = 0.021$), not associated with HIV disease- and treatment-related factors. Two PHIV (14%) and one control (7%) were classified as cognitively impaired ($p = 0.598$).

## Conclusions

Findings indicate treated HIV-infection to be independently associated with lower IQ and poorer cognitive profiles in PHIV children, irrespective of a background of international adoption.

## Introduction

In industrialized as well as in developing countries, cognitive impairment is increasingly recognized as an important concern in children perinatally infected with the human immunodeficiency virus (PHIV) and treated with combination antiretroviral therapy (cART) [1–3]. The severity of HIV disease and immune suppression, reflected by HIV viral load (VL), Center for Disease Control category C diagnoses and lower CD4$^+$ T-cell counts, have been associated with poorer cognitive functioning in PHIV children [3–6].

In countries such as the Netherlands and the UK, the majority of PHIV children was born abroad, with an increasing proportion having been internationally adopted by adoptive parents [7, 8]. Internationally adopted children are prone to be exposed to childhood adversities, including poor health, economic hardship and compromised rearing environment in their family of origin or within institutions, each of which may have jeopardized their cognitive development [9, 10].

Studying cognition PHIV in industrialized countries, by comparing their cognitive performance with an HIV- uninfected control group of which none have been adopted may potentially introduce biases. Research to date tends to focus on HIV-uninfected (or HIV-exposed but HIV-uninfected) control groups that are matched for age, sex, ethnicity and socioeconomic status (SES), rather than adoption status. Previous results from our NOVICE cohort (neurological, cognitive, and visual performance in perinatally HIV-infected children), indicated poorer cognitive functioning in PHIV compared to controls [3, 11]. However, PHIV children in our cohort were more often adopted compared to the HIV- uninfected control group, this precluding us from investigating any effect of adoption on HIV-associated cognitive impairment.

The current study aims to further elucidate the association between treated HIV-infection and cognitive functioning in children, by addressing the background of international adoption. To study this, we expanded the NOVICE cohort with PHIV children and a well matched internationally adopted HIV- uninfected control group and we cross-sectionally compared cognitive functioning between controls.

## Methods

This cross-sectional study was part of the prospective observational NOVICE cohort study investigating the effect of perinatal HIV infection and cART exposure on neurological,

cognitive and visual performances, conducted at the Amsterdam University Medical Centers (AUMC), University of Amsterdam, the Netherlands [3, 12–18]. Among all PHIV children in the outpatient department of our hospital we newly recruited those who were 12 years or older between February 2017 and July 2018[3]. We frequency matched HIV- uninfected controls to PHIV regarding age, sex, ethnicity and socioeconomic status (SES), comparable to the NOV-ICE cohort. We additionally matched for adoption status and, if applicable, region of adoption. We recruited HIV-uninfected children who had been internationally adopted through two government-licensed adoption organizations (Nederlandse Adoptie Stichting and Stichting Kind en Toekomst), since these arrange adoptions from the relevant regions (sub-Saharan Africa). Both organizations used advertisements on their websites and approached to eligible families by email. All participants could speak and understand the Dutch language. We used the following exclusion criteria (as reported by adoptive parents): (non-HIV associated) chronic neurological diseases such as seizure disorders, (history of) intracerebral neoplasms and infections, severe traumatic brain injury (with loss of consciousness longer than 30 minutes), and severe psychiatric disorders. The ethics committee of the Amsterdam University Medical Centers reviewed and approved the study protocol. We obtained written informed consent from all participants older than 12 years and from all parents or legal guardians younger than 18 years if age. This study was registered with the Nederlands Trial Register (ID NL6813).

## Sociodemographic, adoption and HIV- and cART-related characteristics

We collected the following sociodemographic data: age, sex, ethnicity, region of birth, substance use (alcohol, tobacco, cannabis and drug use), education level of (adoptive) parents (scored according to the International Standard Classification of Education [ISCED])[19] and number of parents with a job. ISCED is scored from 0 to 9, ranging from less than primary education to doctoral or equivalent level[19]. We defined SES by parental education and parental occupational status. We assessed the following data regarding adoption: adoption status (yes/no), adoption categorized as special need adoption (yes/no) meaning that a child needs extra care and attention (for example HIV diagnosis), and age at adoption which we defined based on the date of entry into the Netherlands. For PHIV children, we performed laboratory testing of HIV-1 RNA viral load (VL) and $CD4^+$ T-cell count on the same day as the neuropsychological assessment (NPA). In all controls we performed HIV-testing to confirm their HIV- uninfected status. The Dutch HIV Monitoring Foundation provided historical data (age at HIV diagnosis, route of HIV transmission, age at cART initiation duration of cART, time between HIV diagnosis and cART initiation) and data since registration in the Netherlands (AIDS-defining clinical events, Centers for Disease Control and Prevention [CDC] clinical staging, nadir CD4+ T-cell z-score, zenith VL). cART was defined as use of at least three antiretroviral drugs from two or more classes.

## Neuropsychological assessment

One well-trained neuropsychologist (AMtH) who was blinded with respect to the participants' HIV status performed the NPA in all children (S1 Table). NPA included subtests of the Dutch version of the Wechsler Intelligence Scale for Children (WISC)–III (<16 years of age) or the Wechsler Adult Intelligence Scale (WAIS)–III (>16 years of age): Vocabulary, Arithmetic, Block Design, Picture Arrangement, Digit Span, Coding and Symbol search[20, 21]. Furthermore, NPA included the Trail Making Test (TMT) part A and part B[22], the Dutch adaptation of the Rey Auditory Verbal Learning Test (RAVLT) (immediate [sum of trials 1–5] and delayed recall scores and recognition score)[23], and the Beery–Buktenica Developmental Test

of Visual-Motor Integration (Beery VMI)[24]. NPA took 1.5–2 hours, including short breaks to prevent fatigue.

## Data processing

We standardized WISC or WAIS raw subtests into Wechsler norm scores (with a mean of 10 and standard deviation [SD] of 3) and into scale scores (mean 100, SD 15) using age- and sex-adjusted Dutch norm standards from test manuals[20, 21]. We converted Beery VMI raw scores into t-scores (with a mean of 50 and SD of 10)[24], and standardized TMT and RAVLT raw scores into Z-scores, as no appropriate Dutch reference values were available[22, 23, 25]. We calculated Z-scores based on the mean and SD of the control group [25]. We reverse coded variables in order to align interpretation in the same direction: for all tests lower scores indicate poorer test performance. We summarized subtests into the following aggregated domains: intelligence quotient (IQ) working memory, processing speed, learning abilities, executive functioning and visual-motor functioning, based on a principal component analysis (PCA), as described previously [11]. (S1 Table). We summarized all subtests into these cognitive domains to reduce the number of dependent variables (and, thereby to lower the chance of Type I errors) and to draw conclusions on clinically relevant domains.

## Multivariate normative comparison

We used multivariate normative comparison (MNC) to compare complete cognitive profiles to a norm group, by performing one comparison in a multivariate manner [26–28]. This statistical method may be thought of as a multivariate version of Student's t-test for one sample and can be used to statistically compare a complete cognitive profile to the distribution of cognitive profiles of the HIV-uninfected control sample, taking the covariance between all test scores into account, rather than comparing results of individual tests to the control sample[26]. MNC provides a dichotomous result, which indicates whether each participant is classified as cognitively impaired vs not cognitively impaired, and a continuous outcome, reflecting the degree of cognitive deviation of each participant compared to the control sample (represented by the Hotelling's $T^2$). To perform MNC analysis, we included scores of IQ and processing speed, and the individual subtest scores of executive functioning, learning and visual-motor function. We excluded the recognition subtest of learning ability due to the skewed distribution. As only complete cases can be analyzed in MNC, we imputed missings in outcome variables using the mean of the group.

## Statistical analysis

We compared descriptive data for sociodemographic variables using the unpaired t-test for normally distributed data, the Mann-Whitney U test for non-normally distributed data, and the Fischer's exact test for categorical data.

To investigate the association between HIV and cognitive performance we used linear regression models with each cognitive domain as continuous outcome. We adjusted the domains of processing speed, working memory, executive function, learning ability and visual-motor function for IQ in order to prevent possible obscuring of IQ effects on outcomes of other domains [11].

Among PHIV group, we explored associations between cognitive outcomes which significantly differed between groups and the following HIV- and treatment-related characteristics: having history of an AIDS defining disease (Centers for Disease Control and Prevention, category C), nadir CD4+ T-cell z-score, zenith HIV VL and age at cART initiation.

For data acquisition and management, we used OpenClinica software (version 3.6) and we performed statistical analysis using R (R versions 1.1.383)[29]. We considered a two-sided $p < 0.05$ as statistically significant.

## Results

### Participants

We identified nineteen PHIV children from the outpatient department of our hospital, of whom five (28%) declined consent and one (5%) did not meet the inclusion criteria. Thirteen (68%) provided consent to participate. We additionally included one PHIV child, receiving care in another treatment center, whose adoptive parents approached us in response to the advertisements for healthy adopted controls.

Table 1 compares relevant sociodemographic characteristics between PHIV children and HIV- uninfected controls and presents HIV-related characteristics of PHIV children. Participants had a mean age of 10.5 years (SD 1.7) at time of assessment. Thirteen out of the fourteen PHIV children (93%) and twelve out of fifteen HIV- uninfected controls (80%) had a history of international adoption. Groups did not differ, either clinically or statistically, in terms of age, sex, ethnicity, SES of the (adoptive) parents, adoption status, region of birth and age at adoption. The majority of children in both groups (54% in PHIV children and 77% in HIV-uninfected controls) was adopted by a double-income couple with high educational level (85% PHIV and 93% HIV- uninfected controls were living in families with a highest ISCED score of 6–8). Children had been adopted at a mean age of 3.4 years (SD 2.1) and the majority had been born in sub-Saharan Africa. PHIV were diagnosed at a median age of 2.1 years (interquartile range [IQR] 0.3–3.5). All PHIV children were on cART and virologically suppressed at the time of assessment. They had initiated treatment at a median age of 3.07 years old (IQR 1.06–5.41), with a mean duration of 6.8 years (SD 2.6) at the time of assessment.

### Cognitive functioning

Table 2 presents the outcomes of NPA in PHIV and HIV- uninfected children for each separate domain. One PHIV participant (7%) had missing data for the TMT-B subtest of the domain executive functioning and one HIV-uninfected participant (7%) had missing data for the recognition subtest of the RAVLT of the executive functioning domain. For these two participants we based the domain scores on the other subtests of the domain.

Compared to HIV-uninfected participants, PHIV children scored significantly poorer on the domain IQ (unadjusted beta coefficient -15.57, 95% confidence interval [CI] -25.93 to -5.21, $p$-value = 0.005). Moreover, the PHIV group scored lower on working memory (unadjusted beta coefficient: -1.59, 95% CI -3.06 to -0.12, $p$-value = 0.035) and visual-motor function (unadjusted beta coefficient: -8.35, 95% CI -15.10 to -1.59, $p$-value = 0.017). Following adjustment for IQ, both differences were no longer statistically significant (working memory: adjusted beta coefficient -0.65, 95% -2.23 to 0.93, $p$-value = 0.406; and visual-motor function: -3.81, 95% -10.98 to 3.37, $p$-value = 0.286 respectively). The domains of processing speed, executive functioning and learning ability were not significantly different between groups.

### Multivariate normative comparison

Using the MNC to classify cognitive impairment, two PHIV children (14%) and one HIV-uninfected control (7%) were classified as cognitively impaired. The number of children being classified as cognitively impaired did not statistically differ between PHIV children and HIV-uninfected controls ($p$ = 0.598). Using the MNC Hotelling's $T^2$ as a continuous outcome,

**Table 1. Demographics and disease characteristics of Perinatally Human Immunodeficiency Virus (PHIV)-infected (PHIV) and PHIV uninfected (PHIV-) follow-up participants in the NOVICE cohort.**

| | N | PHIV | N | PHIV- | *p*-value |
|---|---|---|---|---|---|
| **Number participants** | | 14 | | 15 | |
| **Male sex, No. (%)** | 14 | 5 (36%) | 15 | 7 (47%) | 0.710 |
| **Age (y), mean (SD)** | 14 | 10.45 (1.73) | 15 | 10.69 (2.79) | 0.785 |
| **Region of birth, No. (%)** | 14 | 1 (7%) | 15 | | 0.242 |
| The Netherlands | | 0(0%) | | 1(7%) | |
| Sub-Saharan Africa | | 13 (93%) | | 10 (67%) | |
| Other | | 1 (7%) | | 4 (27%) | |
| **Ethnicity, No. (%)** | 14 | | 15 | | 0.164 |
| Black | | 13 (93%) | | 11 (73%) | |
| Caucasian | | 0 (0%) | | 2 (13%) | |
| Mixed | | 0 (0%) | | 2 (13%) | |
| Other | | 1 (7%) | | 0 (0%) | |
| **Adoption status, No. (%)** | 14 | | 15 | | 0.598 |
| Not adopted | | 1 (7%) | | 3 (20%) | |
| Adopted | | 13 (93%) | | 12 (80%) | |
| **Age at adoption, median (IQR)** | 13 | 3.3 (2.1–4.2) | 11 | 3.0 (1.2–5.2) | 1 |
| **Special need adoption** | 13 | 13 (100%) | 11 | 2 (18%) | <0.001 |
| **ISCED highest, median (IQR)** | 13 | | 15 | | 0.583 |
| 0–2 | | 2 (18%) | | 1 (7%) | |
| 3–5 | | 0 (0%) | | 0 (0%) | |
| 6–8 | | 11 (85%) | | 14 (93%) | |
| **Number of parents with a job, No. (%)** | 13 | | 13 | | 0.202 |
| 0 | | 0 (0%) | | 1 (8%) | |
| 1 | | 6 (46%) | | 2 (15%) | |
| 2 | | 7 (54%) | | 10 (77%) | |
| **Lifestyle, No. (%)** | | | | | |
| Ever smoked | 14 | 0 (0%) | 15 | 0 (0%) | 1 |
| Ever used alcohol | 14 | 0 (0%) | 15 | 0 (0%) | 1 |
| Ever used recreational drugs | 14 | 0 (0%) | 15 | 0 (0%) | 1 |
| **IQ** | 14 | 81.43 (11.44) | 15 | 97.00 (15.31) | 0.005 |
| **Age at HIV diagnosis (y), median (IQR)** | 14 | 2.10 (0.33–3.45) | | - | |
| **Perinatal transmission, No. (%)** | 14 | 14 (100%) | | - | |
| CDC category*, No. (%) | 14 | | | – | |
| N/A | | 12 (86%) | | | |
| B | | 1 (7%) | | | |
| C | | 1 (7%) | | | |
| Nadir CD4+ T-cell z-score*, median (IQR) | 14 | -0.76 (-1.35 to -0.13) | | - | |
| Zenith HIV viral load (copies/mL)*, median (IQR) | 14 | 107492 (265–296408) | | - | |
| **cART, No. (%)** | 14 | 14 (100%) | | - | |
| **Age at cART initiation (y), median (IQR)** | 14 | 3.07 (1.06–5.41) | | - | |
| **Duration of cART use (y), mean (SD)** | 14 | 6.79 (2.56) | | - | |
| **HIV diagnosis to cART initiation (mo), median (IQR)** | 14 | 22.92 (4.00–59.38) | | - | |
| **Undetectable HIV VL at assessment, No. (%)** | 14 | 14 (100%) | | - | |

Undetectable is defined as HIV RNA < 40c/mL, and allowing viral blips. Values are reported as mean (SD), median (IQR) or n (%). Abbreviations: n, number; y, year; mo, month; m, meter; kg, kilogram; ISCED, International Standard Classification of Education; CDC, Centers for Disease Control and Prevention; N, nonsymptomatic; cART, combination antiretroviral therapy; mo, months.

* Since registration in the Netherlands.

**Table 2. Cognitive domain scores in Perinatally Human Immunodeficiency Virus infected (PHIV) and PHIV uninfected (PHIV-) controls.**

|  | N | PHIV | N | PHIV- | Unadjusted | *p-value* | Adjusted | *p-value* |
|---|---|---|---|---|---|---|---|---|
|  |  | Coefficient (95% CI) |  | Coefficient (95% CI) | Coefficient (95% CI) |  | Coefficient (95% CI) |  |
| IQ | 14 | 81.43 (11.44) | 15 | 97.00 (15.31) | -15.57 (-25.93 to -5.21) | 0.005 | - |  |
| Processing speed | 14 | 104.21 (11.98) | 15 | 104.53 (10.4) | -0.32 (-8.86 to 8.22) | 0.939 | 5.46 (-3.59 to 14.51) | 0.226 |
| Working memory | 14 | 9.14 (1.92) | 15 | 10.73 (1.94) | -1.59 (-3.06 to -0.12) | 0.035 | -0.65 (-2.23 to 0.93) | 0.406 |
| Executive functioning | 14 | -0.20 (0.73) | 15 | 0.00 (0.88) | -0.20 (-0.83 to 0.44) | 0.530 | -0.00 (-0.72 to 0.72) | 0.992 |
| Learning ability | 14 | -0.32 (0.53) | 15 | 0.00 (0.95) | -0.32 (-0.91 to 0.27) | 0.277 | -0.37 (-1.08 to 0.32) | 0.279 |
| Visual-motor function | 14 | 39.79 (10.86) | 15 | 48.13 (6.47) | -8.35 (-15.10 to -1.59) | 0.017 | -3.81 (-10.98 to 3.37) | 0.286 |

Absolute scores and regression coefficients are shown, including 95% CI and *p*-value. IQ and processing speed are reported in Wechsler scale scores (mean 100, SD 15), working memory in Wechsler norm scores (mean 10, SD 3). Executive functioning and learning ability are reported in Z-scores based on the control group (mean 0, SD 1) and visual-motor function are reported in t-scores (mean 50, SD 10). Lower scores indicate worse test performance. Abbreviations: CI, confidence interval; IQ, intelligence quotient.

PHIV children ranged between -13.03 and 1.426, with a median of -3.75 and a mean of -4.36. The HIV-uninfected controls ranged between -12.10–7.71, with a median of 1.37 and a mean of 0.16 ($p = 0.021$).

## Determinants of IQ and cognitive profile by multivariate normative comparison

Among PHIV children, we explored HIV- and cART related determinants of IQ and cognitive functioning measured by MNC (Table 3). We did not find an association between HIV-disease severity markers, such as CD4[+] T-cell, log HIV VL zenith and having an AIDS diagnosis, and IQ and cognitive profile by MNC ($p>0.477$), nor did we find an association between cART-related factors, such as age at cART initiation and time between diagnosis and cART initiation and intellectual and cognitive profile by MNC (all $p>0.380$).

## Discussion

In this cross-sectional study, we aimed to elucidate whether previous found poorer cognition in PHIV children could have been confounded by having a background of international adoption. To adjust for confounding factors that potentially converge with having a background of international adoption, we compared cognitive performance between PHIV children and a

**Table 3. Associations between HIV- and cART-related variables and intellectual performance in PHIV children.**

|  |  | IQ |  | MNC continuous outcome |  |
|---|---|---|---|---|---|
|  | N | Univariate (95% CI) | *p-value* | Univariate (95% CI) | *p-value* |
| **CDC-C**[*] | 14 | -3.692 (-30.5–23.1) | 0.769 | -3.523 (-14.0–6.9) | 0.477 |
| **HIV VL zenith (log copies/mL)**[*] | 14 | -0.190 (-4.4–4.1) | 0.924 | 0.011 (-1.7–1.7) | 0.989 |
| **CD4+ T-cell nadir (z-score)**[*] | 14 | -1.060 (-10.1–7.9) | 0.818 | 1.132 (-2.4–4.6) | 0.496 |
| **Age at start cART (years)** | 14 | -0.894 (-3.0–1.2) | 0.380 | 0.350 (-3.0–1.2) | 0.386 |
| **Time between diagnoses and start cART (years)** | 14 | -0.769 (-2.7–1.1) | 0.395 | -0.276 (-1.0–0.5) | 0.443 |

No variables were included in multivariable regression analysis since none of the variables had a *p*-value < .20 in univariable analysis. Abbreviations: IQ, intelligence quotient; N, number; CDC-C, Centers for Disease Control and Prevention, category C; CI, confidence interval; HIV, Human Immunodeficiency Virus; VL, Viral Load; cART, combination antiretroviral therapy.

[*]Registered since registration in the Netherlands.

well matched HIV-uninfected control group–being similar in having a background of international adoption. Results of this study suggest significantly lower IQ in PHIV children, but no specific cognitive deficiency in additional domains. As these observations are done in a well-controlled study, we may conclude that the observed lower IQ in our cohort of perinatally infected children with suppressed HIV infection on treatment is independent of factors related to international adoption.

Poorer intellectual performance in PHIV children compared to adoption matched peers is consistent with previous evidence from the NOVICE cohort using non-adopted control group [3, 11]. Both cross-sectional and longitudinal evidence from the NOVICE cohort, showed poorer intellectual functioning in PHIV children compared to both Dutch norm group and controls matched for age, sex, ethnicity and SES [3, 11]. The persistent finding of lower IQ in PHIV children, also when matching for adoption status, further supports the association between treated HIV and IQ.

The findings of the current study differ, however, from a recent German study, reporting on IQ in fourteen PHIV children with a median age of 8.2 years old to be not statistically different from the norm, indicating normal intellectual development [30]. Further, they found IQ to be significantly inversely associated with the initiation of cART within the first year of life, warranting early initiation. A possible explanation for the discrepancy with our results might be due to the lack of children in the current study who had initiated cART within the first year of life, lacking power to detect an association.

Longitudinal observation of the NOVICE cohort showed similar trajectories in overall cognitive performance over time compared to healthy peers, yet suggested executive dysfunctioning in PHIV children once having reached adolescence[11]. However in this cross-sectional design–with regards to other cognitive domains–our results suggest no differences between groups in the domains of processing speed, working memory, executive function, learning ability and visual motor function. Unadjusted models however showed significant differences between groups in the domains working memory and visual motor function. A possible explanation for this might be that the association between HIV-infection and both working memory and visual motor function in fact reflects difference in IQ, or there might have been too little power to detect actual significant associations. With regard to executive function, however, delay can still become apparent during adolescence, as we showed earlier [11].

Comparing complete cognitive profiles between groups using multivariate normative comparison[26], our findings indicate a clinical and significant poorer overall cognitive profile in PHIV children compared to healthy controls. Clinically, we found a higher prevalence of cognitive impairment in PHIV children compared to healthy controls, being not statistically significant. Significant differences in cognitive impairment are consistent with those of previous results from the NOVICE cohort study [11].

We found no significant associations between IQ and HIV- or cART-related factors among the PHIV group. It seems possible that these results are due to a lack of data regarding the severity of HIV infection in the children before entering the Netherlands.

Interestingly, we found that the HIV-uninfected controls included in the current study scored on average 10 IQ points higher (95% CI 0.83 to 18.17) compared to the controls of the previous NOVICE assessment, in whom the prevalence of international adoption background was much lower[3]. Higher IQ in the current HIV-uninfected controls might be attributed to the enriched environment of adoptive families, being mostly well above average socioeconomically. The Flynn effect, describing the worldwide increase in intelligence test scores over time, might also have contributed to this[31].

In industrialized countries, no studies are yet available that acknowledge a background of international adoption as a potential confounder when studying the association between

treated HIV-infection and cognition. In the event of international adoption, children have often been exposed to adverse conditions in early life, such as adverse prenatal and/or postnatal rearing conditions, loss of biological parents and residing in an institutional environment during early life. These factors all potentially impact normal cognitive development.

Despite being the first study investigating treated HIV-associated cognitive performance with the inclusion of a well-matched internationally adopted control group, a number of limitations needs to be considered. As for any study of this kind, we cannot rule out the occurrence of selection bias. Possible selection bias might have occurred due to a difference in reasons for relinquishment or abandonment across groups, which is partly reflected by the variation of special need adoption across groups and in the Netherlands [32]. Further, selection bias might have been introduced unintentionally due to selective inclusion of healthy controls through adoption organizations. Although we were unaware of any specific selection bias in the recruitment of controls, we were unable to formally assess this.

The study is limited by the lack of information on early life conditions, such as adverse institutional rearing environment, prematurity, and pregnancy conditions [33]. Therefore, we were unable to analyse these variables. Although we matched groups for (adoptive) parental SES, we were unable to take into consideration genetic factors as a potential confounder in cognitive development. Evidence from longitudinal studies show that IQ of adopted children becomes more similar to the IQ of their birth parents with increasing age, indicating a possible progressive influence of genetic factors as these children age[34].

Since this was a single center study with a limited study sample, this study may have been underpowered to detect smaller differences. Due to the low sample size, we were unable to perform extensive multivariable regression analyses or subgroup analyses. Both significant and non-significant results should be interpreted with caution and results should be replicated by larger (preferably longitudinal) studies[35]. Due to the cross-sectional design of this study we are unable to infer causality. Longitudinal studies including PHIV children and internationally adopted controls would be valuable, establish temporality, which is one of the criteria for determining causal inference.

Taken this all together, the results of this study suggest treated HIV-infection to be independently associated with lower IQ and a poorer cognitive profile in PHIV children, irrespective of a background of international adoption.

## Supporting information

**S1 Table. Neuropsychological assessment.** Abbreviations: IQ, intelligence quotient, WISC, Wechsler Intelligence Scale for Children; WAIS, Wechsler Adult Intelligence Scale. *We obtained domain scores on learning ability and executive functioning by averaging the subtest scores.
(DOCX)

## Acknowledgments

We would like to thank all study participants and their parents for participating, Annouschka Weijsenfeld and Claudia de Boer, HIV pediatric nurses, for their support with the recruitment of patients at the outpatient department. We acknowledge the contribution of Em. Prof. Dr. B. A. Schmand and Prof. Dr. H.M. Huizenga for their guidance using the MNC method, and Prof. dr. F. Juffer for her expertise and helpful discussion to familiarize us in adoption research.

## Author Contributions

**Conceptualization:** D. Pajkrt.

**Data curation:** M. Van den Hof.

**Formal analysis:** M. Van den Hof, A. M. ter Haar.

**Funding acquisition:** D. Pajkrt.

**Investigation:** M. Van den Hof, A. M. ter Haar, H. J. Scherpbier.

**Methodology:** M. Van den Hof, F. W. N. M. Wit, K. J. Oostrom.

**Project administration:** M. Van den Hof.

**Supervision:** P. Reiss, D. Pajkrt.

**Writing – original draft:** M. Van den Hof.

**Writing – review & editing:** A. M. ter Haar, H. J. Scherpbier, P. Reiss, F. W. N. M. Wit, K. J. Oostrom, D. Pajkrt.

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
