## [Decision Letter · Decision Letter 0]

10 Sep 2019

PONE-D-19-18957

Lower IQ and poorer cognitive profiles in treated perinatally HIV-infected children is irrespective of having a background of international adoption

PLOS ONE

Dear MD Van den Hof,

Thank you for submitting your manuscript to PLOS ONE. After careful consideration by 2 Reviewers and an Academic Editor, all of the critiques of both Reviewers must be addressed in detail in a revision to determine publication status. If you are prepared to undertake the work required, I would be pleased to reconsider my decision, but revision of the original submission without directly addressing the critiques of the two Reviewers does not guarantee acceptance for publication in PLOS ONE. If the authors do not feel that the queries can be addressed, please consider submitting to another publication medium. A revised submission will be sent out for re-review. The authors are urged to have the manuscript given a hard copyedit for syntax and grammar.

**Comments to the Author**

1. Is the manuscript technically sound, and do the data support the conclusions?

Reviewer #1: Partly

Reviewer #2: Yes

2. Has the statistical analysis been performed appropriately and rigorously? 

Reviewer #1: I Don't Know

Reviewer #2: Yes

3. Have the authors made all data underlying the findings in their manuscript fully available?

Reviewer #1: Yes

Reviewer #2: No

4. Is the manuscript presented in an intelligible fashion and written in standard English?

Reviewer #1: Yes

Reviewer #2: Yes

5. Review Comments to the Author

Reviewer #1: The authors present neurocognitive data on a small cohort of HIV + and HIV neg children/young teens. They note a significant difference in IQ between the 2 groups (who were fairly well matched between groups in terms of age, area of adoption, etc). The manuscript is well written, the data is presented clearly, and the statistical analysis seems appropriate.

The manuscript suffers from two main issues:

1.The sample size is very small for a study of neurocognitive outcomes between two groups-the difference in IQ was still significant (based on how large the difference was) but the sample size may well have limited other associations.

2 It is unclear why the authors chose to include non adopted subjects (1 HIV+ and 3 HIV negative were not adopted). It would seem that the authors should have been presented data that just included the adopted subjects, or, at the very least, presented a secondary analysis done on the smaller group that was adopted. They do not explain in the manuscript the reasoning behind the non adopted subjects inclusion.

The difference in cognitive scores parallels studies done in the early, pre-cART era. It should be noted that in studies from the United States, when comparing adolescents with perinatally acquired HIV vs HIV exposed but uninfected adolescents, have found near equal overall IQ between the two groups, but significantly lower than national norms (ie, in the 80s, instead of averaging 100), which for the US suggests family or social/environmental issues played a larger role on determining cognitive outcomes than HIV status (in the era of cART).

As the authors note, for the HIV positive subjects,lacking clinical and HIV-related information about their early few years, prior to adoption, and small sample size, makes it hard to tease out the determinants of the lower IQs.

Reviewer #2: This is an interesting study that examines cognitive functioning among perinatally HIV-infected (PHIV+) children who were born abroad and adopted in the Netherlands and compared to HIV-negative Dutch adoptees of foreign birth. This is the first study to compare cognitive functioning across demographically similar (including country of birth) adoptees - an important addition to the literature as cognitive tests can suffer from biases that may impact test scores when examinees are not compared to truly similar populations. While this study will make an important contribution to the study, its impact is tempered as the sample size is very small (though it is understood that matching adoptees to country/region of birth is very challenging and limits sample size). There are a few other concerns.

1) In the Introduction, the authors state that in industrialized countries, the majority of PHIV+ children were born abroad and adopted. This may be the case in the UK and the Netherlands, but these two countries are not the majority of industrialized countries. In the US, there is a substantial population of PHIV+ people who were born there.

2) It’s not clear how impairment was defined using the multivariate normative comparison (MNC) approach. Is there a threshold of covariance that determines impairment or a standard deviation below (or above) a certain amount?

3) While the MNC approach does not compare individual test performance, it could be important to know if there are certain tests that show more impaired performance than others.

6. PLOS authors have the option to publish the peer review history of their article (what does this mean?). If published, this will include your full peer review and any attached files.

**Do you want your identity to be public for this peer review?** For information about this choice, including consent withdrawal, please see our Privacy Policy.

Reviewer #1: No

Reviewer #2: No

We would appreciate receiving your revised manuscript by March, 2020. To enhance the reproducibility of your results, we recommend that if applicable you deposit your laboratory protocols in protocols.io, where a protocol can be assigned its own identifier (DOI) such that it can be cited independently in the future. For instructions see: http://journals.plos.org/plosone/s/submission-guidelines#loc-laboratory-protocols

We look forward to receiving your revised manuscript.

Kind regards,

Stephen D. Ginsberg, Ph.D.

Section Editor

PLOS ONE

Journal Requirements:

2. You indicated that you had ethical approval for your study. In your Methods section, please ensure you have also stated whether you obtained consent from parents or guardians of the minors included in the study or whether the research ethics committee or IRB specifically waived the need for their consent

---

## [Author Response · Author response to Decision Letter 0]

2 Oct 2019

Comments to the reviewers’ comments

REVIEWER COMMENTS

Reviewer #1: The authors present neurocognitive data on a small cohort of HIV+ and HIV neg children/young teens. They note a significant difference in IQ between the 2 groups (who were fairly well matched between groups in terms of age, area of adoption, etc). The manuscript is well written, the data is presented clearly, and the statistical analysis seems appropriate.

We sincerely thank reviewer 1 for the evaluation and appreciation of our study. We will address the specific questions and remarks below. 

The manuscript suffers from two main issues:

1. The sample size is very small for a study of neurocognitive outcomes between two groups-the difference in IQ was still significant (based on how large the difference was) but the sample size may well have limited other associations.

Response: we fully agree with the reviewers comment. To address this limitation we have acknowledged this in the discussion section as follows (paragraph 10): 

“Since this was a single center study with a limited study sample, this study may have been underpowered to detect smaller differences. Due to the low sample size, we were unable to perform extensive multivariable regression analyses or subgroup analyses. Both significant and non-significant results should be interpreted with caution and results should be replicated by larger (preferably longitudinal) studies”.

2. It is unclear why the authors chose to include non-adopted subjects (1 HIV+ and 3 HIV negative were not adopted). It would seem that the authors should have been presented data that just included the adopted subjects, or, at the very least, presented a secondary analysis done on the smaller group that was adopted. They do not explain in the manuscript the reasoning behind the non-adopted subjects’ inclusion.

Response: we thank the reviewer for this comment. Over the last decade, the demographics of the population of PHIV has changed considerably, with an increase of the proportion of PHIV-infected children with a background of international adoption. The main aim of this manuscript was to investigate neurocognitive performance in PHIV-infected children growing into adulthood. Therefore we aimed to include a matched HIV-uninfected control group with children who have been exposed to international adoption as well, in order to match HIV-uninfected controls as best as possible, in order to account for unknown confounders such as childhood adversities, poor health, compromised rearing environment etc. Since not all PHIV-infected children have been adopted (1 HIV+), we also included non-adopted children in the HIV-uninfected group. In the methods section, we have tried to clarify the matching procedure as follows: “We frequency matched HIV-negative controls to PHIV regarding…”.

The difference in cognitive scores parallels studies done in the early, pre-cART era. It should be noted that in studies from the United States, when comparing adolescents with perinatally acquired HIV vs HIV exposed but uninfected adolescents, have found near equal overall IQ between the two groups, but significantly lower than national norms (ie, in the 80s, instead of averaging 100), which for the US suggests family or social/environmental issues played a larger role on determining cognitive outcomes than HIV status (in the era of cART).

As the authors note, for the HIV positive subjects, lacking clinical and HIV-related information about their early few years, prior to adoption, and small sample size, makes it hard to tease out the determinants of the lower IQs.

Response: we agree with the reviewer on this comment. In children with a background of international adoption, often, there is a lack of information on early life conditions and early HIV infection, as determinants for cognitive performance. We believe, however, that comparing adopted children to a group of children that also has experienced international adoption, we minimize the confounders that coincide with international adoption, such as the exposure to childhood adversities, poor health, compromised rearing environment etc. We do acknowledge that we cannot rule out the existence of potential residual confounders completely.

Reviewer #2: This is an interesting study that examines cognitive functioning among perinatally HIV-infected (PHIV+) children who were born abroad and adopted in the Netherlands and compared to HIV-negative Dutch adoptees of foreign birth. This is the first study to compare cognitive functioning across demographically similar (including country of birth) adoptees - an important addition to the literature as cognitive tests can suffer from biases that may impact test scores when examinees are not compared to truly similar populations. While this study will make an important contribution to the study, its impact is tempered as the sample size is very small (though it is understood that matching adoptees to country/region of birth is very challenging and limits sample size). There are a few other concerns.

1. In the Introduction, the authors state that in industrialized countries, the majority of PHIV+ children were born abroad and adopted. This may be the case in the UK and the Netherlands, but these two countries are not the majority of industrialized countries. In the US, there is a substantial population of PHIV+ people who were born there.

Response: we thank the reviewer for this remark. We have changed the manuscript as follows: “In countries such as the Netherlands and the UK…”

2) It’s not clear how impairment was defined using the multivariate normative comparison (MNC) approach. Is there a threshold of covariance that determines impairment or a standard deviation below (or above) a certain amount?

Response: we thank the reviewer for this question. The MNC method can be used to statistically compare multiple cognitive scores of each single PHIV participant against the distributions of the cognitive profile of the HIV-uninfected control group as a whole, taking the covariance between all test scores into account. The test statistic Hotelling’s T2 is calculated and if the statistic exceeds the critical F value associated with alpha 0.05, the individual participant significantly differs from the norm. For a deeper understanding of the MNC approach, in the manuscript we have referred to the paper of H.M. Huizenga et al (Huizenga HM, Smeding H, Grasman RP, Schmand B. Multivariate normative comparisons. Neuropsychologia. 2007;45(11):2534-42).

3) While the MNC approach does not compare individual test performance, it could be important to know if there are certain tests that show more impaired performance than others.

Response: we thank the review for this remark. We agree with the reviewer that it could be important to know more about the specifics of the impairment. However, we believe that is more clinically relevant to study impairment on a cognitive domain level instead of at the level of individual tests. Furthermore, summarizing cognitive tests into domains lowers the chance of Type I error. We investigate potential impairment in specific domains by comparing mean (Z-) scores of cognitive domains (intelligence quotient [IQ] working memory, processing speed, learning abilities, executive functioning and visual-motor functioning). 

We clarified on this as follows (Method, data processing): “We summarized all subtests into these cognitive domains to reduce the number of dependent variables (and, thereby to lower the chance of Type I errors) and to draw conclusions on clinically relevant domains.”

• A rebuttal letter that responds to each point raised by the academic editor and reviewer(s). This letter should be uploaded as separate file and labeled 'Response to Reviewers'.

• A marked-up copy of your manuscript that highlights changes made to the original version. This file should be uploaded as separate file and labeled 'Revised Manuscript with Track Changes'.

• An unmarked version of your revised paper without tracked changes. This file should be uploaded as separate file and labeled 'Manuscript'.

We look forward to receiving your revised manuscript.

Kind regards,

Stephen D. Ginsberg, Ph.D.

Section Editor

PLOS ONE

Journal Requirements:

Thank you for this comment. We have changed the manuscripts style to meet PLOS ONE’s requirements. 

2. You indicated that you had ethical approval for your study. In your Methods section, please ensure you have also stated whether you obtained consent from parents or guardians of the minors included in the study or whether the research ethics committee or IRB specifically waived the need for their consent

Thank you for this comment. Indeed, we obtained informed consent from parent/guardians of the minors. To clarify this, we added this in the Methods section, as follows:

“We obtained written informed consent from all participants older than 12 years and from all parents or legal guardians younger than 18 years if age”.

---

## [Decision Letter · Decision Letter 1]

25 Oct 2019

Lower IQ and poorer cognitive profiles in treated perinatally HIV-infected children is irrespective of having a background of international adoption

PONE-D-19-18957R1

Dear Dr. Van den Hof,

We are pleased to inform you that your manuscript has been judged scientifically suitable for publication and will be formally accepted for publication once it complies with all outstanding technical requirements.

With kind regards,

Stephen D. Ginsberg, Ph.D.

Section Editor

PLOS ONE

**Comments to the Author**

1. If the authors have adequately addressed your comments raised in a previous round of review and you feel that this manuscript is now acceptable for publication, you may indicate that here to bypass the “Comments to the Author” section, enter your conflict of interest statement in the “Confidential to Editor” section, and submit your "Accept" recommendation.

Reviewer #1: All comments have been addressed

2. Is the manuscript technically sound, and do the data support the conclusions?

Reviewer #1: (No Response)

3. Has the statistical analysis been performed appropriately and rigorously? 

Reviewer #1: (No Response)

4. Have the authors made all data underlying the findings in their manuscript fully available?

Reviewer #1: (No Response)

5. Is the manuscript presented in an intelligible fashion and written in standard English?

Reviewer #1: (No Response)

6. Review Comments to the Author

Reviewer #1: (No Response)

7. PLOS authors have the option to publish the peer review history of their article (what does this mean?). If published, this will include your full peer review and any attached files.

Reviewer #1: No

---

## [Editor Report · Acceptance letter]

20 Nov 2019

PONE-D-19-18957R1 

Lower IQ and poorer cognitive profiles in treated perinatally HIV-infected children is irrespective of having a background of international adoption 

Dear Dr. Van den Hof:

I am pleased to inform you that your manuscript has been deemed suitable for publication in PLOS ONE. Congratulations! Your manuscript is now with our production department. 

With kind regards,

on behalf of

Dr. Stephen D Ginsberg 

Section Editor

PLOS ONE